# Lumpy Skin Disease Virus ORF137 Protein Inhibits Type I Interferon Production by Interacting with and Decreasing the Phosphorylation of IRF3

**DOI:** 10.3390/cells14181475

**Published:** 2025-09-22

**Authors:** Qunhua Ke, Kaishen Yao, Min Qu, Zhengji Liang, Miaomiao Li, Xiangwei Wang, Xiangping Yin, Yuefeng Sun

**Affiliations:** 1College of Veterinary Medicine, Gansu Agricultural University, Lanzhou 730070, China; kqh1298505845@163.com; 2State Key Laboratory for Animal Disease Control and Prevention, College of Veterinary Medicine, Lanzhou University, Lanzhou Veterinary Research Institute, Chinese Academy of Agricultural Sciences, Lanzhou 730046, China; yaokaishen0802@163.com (K.Y.); 13195613233@163.com (M.Q.); zhengjiliang1998@163.com (Z.L.); limiaomiao@caas.cn (M.L.); wangxiangwei@caas.cn (X.W.); 3Yazhouwan National Laboratory, Sanya 572024, China

**Keywords:** lumpy skin disease virus, ORF137, IFN-β, Phospho-IRF3, immune evasion

## Abstract

Lumpy skin disease (LSD) is an invasive infectious disease caused by the lumpy skin disease virus (LSDV), which is detrimental to the production of cattle. LSDV encodes about 156 proteins, most of whose functions are still unknown. In this study, we found that the ORF137 protein was identified as one of the strongest inhibitors of IFN-β and ISG expression, determining LSDV ORF137 as a negative regulator of interferon (IFN) β signaling. Further evidence suggests that ORF137 interacts with the signal transduction factor IRF3 and inhibits the activation of IFN-β signaling by reducing Phospho-IRF3 (p-IRF3). Further investigation indicated that overexpression of ORF137 in BMEC could significantly inhibit the transcription of IFN-β and ISGs, thereby promoting the replication of LSDV. More importantly, through homologous recombination, we deleted the ORF137 gene from the LSDV/FJ/CHA/2021 strain and constructed the recombinant strain LSDV-ΔORF137-EGFP. Compared with the parental strain, LSDV-ΔORF137-EGFP showed a weakened effect on inhibiting the transcription of IFN-β and ISGs and a reduced replication level in infected MDBK cells. In summary, ORF137 facilitates LSDV replication by targeting IRF3 to inhibit IFN-β signaling. Our findings reveal a new mechanism by which LSDV suppresses the host antiviral response, which may facilitate the development of attenuated live vaccines for LSDV.

## 1. Introduction

Lumpy skin disease (LSD) is caused by the lumpy skin disease virus (LSDV), which belongs to the Capripoxvirus (CaPV) genus within the Poxviridae family [1]. The LSDV genome consists of a double-stranded DNA (dsDNA) genome approximately 151 kb long and contains 156 open reading frames (ORFs) and has a nucleotide identity of over 96% with the genomes of the other members of the CaPV genus: goatpox virus (GTPV) and sheeppox virus (SPPV) [2,3]. The main clinical features of cattle infected with LSDV are fever, emaciation, enlarged lymph nodes, skin edema, and extensive nodular lesions on the skin and mucosal surfaces, which often occur on the neck, back, perineum, and breast, causing significant economic and ecological consequences for the development of the global cattle industry [4,5]. LSD was first discovered in Zambia in 1929 and mostly circulated in Africa until 1986, when it emerged in Israel, gradually spreading to the Middle East, Eastern Europe, Russia, and Asian countries, underscoring the importance of extensive research in LSDV virology and the implementation of more effective biosecurity measures [6,7]. The first case of LSDV infection in China was reported in Yili, Xinjiang, China in August 2019, and it began circulating in several provinces shortly thereafter [8,9]. Because the functions of most of LSDV encoding proteins are unknown, the molecular mechanisms of its pathogenesis and immune escape have not been fully studied, so there are no safe and effective commercial LSDV vaccine and targeted therapeutic drugs at present [2,6,10].

The innate immune response is of prime importance in the immediate recognition and elimination of invading micro-organisms [11]. Pathogen-associated molecular pattern molecules (PAMPs) are derived from microorganisms and recognized by pattern recognition receptor (PRR)-bearing cells of the innate immune system as well as many epithelial cells [12,13]. Circular GMP-AMP synthetase (cGAS) is a PRR that recognizes DNA in the cytoplasm. When combined with double-stranded DNA (dsDNA), cGAS is activated and synthesized from the cyclic dinucleotide 2′3′-cGAMP as a second messenger for the binding to signal adapter interferon gene stimulating factor (STING). Activated STING leads to the activation of TANK-bound kinase 1 (TBK-1), which further phosphorylates TBK-1 and activates the transcription factor interferon regulator factor 3 (IRF3), thereby inducing the transcription and expression of type I IFN and its stimulating genes (ISGs) [14]. These genes directly affect protein synthesis, cell growth, and survival, thus establishing an antiviral state [13].

Although genetic and functional studies suggest that LSDV has multiple immunomodulatory genes that can exert an innate immune response early in infection, the underlying molecular mechanisms are poorly understood [2,15]. It has been demonstrated that LSDV can suppress the host innate immune response through the following pathway. For example, LSDV ORF127 is a putative EEV glycoprotein; this protein interacts with TBK-1 located in the cytoplasm and inhibits K63-linked polyubiquitination of TBK-1, ultimately attenuating TBK-1-mediated interferon production and downregulating the antiviral immune response [16]. The absence of LSDV ORF012 (an ankyrin) significantly affects viral replication, and further studies show that LSDV ORF012 interacts with interferon-induced proteins with tetrapeptide repeats (IFIT), especially IFIT 1, changing its subcellular localization, interacting with its C terminus, and inhibiting its RNA binding ability, highlighting the importance of LSDV012 in antagonizing type I IFN responses [17]. The 001 protein of lumpy skin disease virus (LSDV) is a virulence factor. LSDV001 enhances the inflammatory response by interacting with TAK1 (Transforming Growth Factor-β-Activated Kinase 1) and TAB2/3 (TAK1-Binding Protein 2/3) to promote the formation of the TAK1-TAB2/3 complex [18]. Additionally, other studies have shown that LSDV 001 acts as a negative regulator of the interferon (IFN) signaling pathway by impairing the dimerization of IRF3 (Interferon Regulatory Factor 3) [19]. Furthermore, it has been demonstrated that proteins of multiple DNA viruses can suppress the host innate immune response via the cGAS-STING pathway. The C6 protein of vaccinia virus (VACV) contributes to the virulence by inhibiting PRR-induced activation of IRF3 and IRF7 [20]. The B175L protein of African swine fever virus (ASFV) and the ORF 48 protein encoded by Kaposi’s sarcoma-associated herpesvirus (KSHV) interact with key nodal molecules in the antiviral signaling pathway mediated by cGAS-STING in the host innate immune system, thereby achieving immune evasion and ultimately helping the viruses establish persistent infections [21,22].

In this study, we aimed to identify the LSDV proteins that inhibit the cGAS-STING pathway. We found that LSDV ORF137 is an important inhibitor of the cGAS-STING antiviral sensing pathway. Mechanistically, LSDV ORF137 interacts with and decreases the phosphorylation of IRF3, leading to reduced interferon expression, and ultimately downregulates the antiviral immune response. These data illustrate the ORF137 protein-mediated suppression of host antiviral defense and provide new insights into the pathogenesis of LSDV and vaccine development.

## 2. Materials and Methods

### 2.1. Cells and Viruses

Human embryonic kidney 293T (HEK-293T, EDJ-WQ0466, Guangzhou Editgene Co., Ltd., Guangzhou, China) cells, Mardin Darby Bovine Kidney (MDBK, EDJ-WQ0762, Guangzhou Editgene Co., Ltd., Guangzhou, China) cells, and bovine mammary epithelial cells (BMEC, MZ-2690, Mingzhou biotechnology co. ltd, Ningbo, China) were cultured in Dulbecco’s modified Eagle’s medium (DMEM; BBI, E600003-0500) supplemented with 10% fetal bovine serum (FBS; CELL-BOX, AUS-01S-02, Changsha, China) and 1% penicillin–streptomycin (Shanghai Epizyme Biomedical Technology Co., Ltd., CB010, Shanghai, China) at 37 °C with 5% CO_2_. LSDV/FJ/CHA/2021 strain (GenBank: OP752701) was isolated and stored in the biosafety level 3 (BSL-3) laboratory of Lanzhou Veterinary Research Institute Chinese Academy of Agricultural Sciences (LVRI CAAS), and the LSDVΔORF137-EGFP strain was constructed and purified by our laboratory. LSDV/FJ/CHA/2021 is denoted by LSDV or WT-LSDV in the text. Herpes simplex virus type 1 (HSV-1) and herpes simplex virus 1 expressing GFP (HSV-GFP) are stored in our laboratory. All LSDV-related experiments were conducted in the BSL-3 laboratory of LVRI. All the aforementioned cells are preserved in our laboratory.

### 2.2. Antibodies and Reagents

#### 2.2.1. Antibodies

All purchased antibodies are in Table 1.

#### 2.2.2. Reagents

The double-luciferase reporter assay kits were purchased from Vazyme (DL101-01, Nanjing, China). Kits for rapid extraction of plasmid DNA were purchased from TIANGEN (DP118-02, Beijing, China). Total RNA Extractor (Trizol) was purchased from Shengong Biotech (B511311-0500, Shanghai, China). Polyvinylidene fluoride (PVDF) membranes were purchased from Cytiva (10600023, Darmstadt, Germany). Transfection reagent lipo8000™ was purchased from Beyotime biotechnology (C0533-7.5 ml, Shanghai, China). Opti-MEM was purchased from Thermo Fisher Scientific (31985070, MA, USA). StarScript II RT Mix with gDNA Remover kit and 2×RealStar Power SYBR qPCR Mix were purchased from Genstar (A234-10, A301-05, Beijing, China).

### 2.3. Construction and Transfection of Plasmids

The 156 eukaryotic expression plasmids expressing LSDV-encoding proteins were synthesized by Wuhan Jinkairui Biotechnology Co., Ltd. (Wuhan, China). The ORF137 gene of LSDV/FJ/CHA/2021 was cloned into the pCAGGS-Flag vector with *EcoR I* and *Kpn I* enzyme digestion; in addition, the ORF137 gene of LSDV/FJ/CHA/2021 was cloned into the pCMV-HA vector with *EcoR I* and *Kpn I* enzyme digestion by GENEWIZ Suzhou, China. The experimental principle of construction followed standard molecular biology techniques. Hs-cGAS-HA, hs-STING-HA, hs-TBK1-HA, hs-IRF3-HA, Bos taurus-TBK1-HA, Bos taurus-IRF3-HA, IFN-β-Luc, pRL-TK, pCAGGS-Flag, and PCMV-HA expression plasmids were stored in our laboratory previously; Bos taurus-cGAS-Flag, Bos taurus-STING-Flag, and the CMV-Flag vector were purchased from miaolingbio.Inc. (Wuhan, China). These plasmids were dissolved in elution buffer and stored at −20 °C. According to the transfection protocol, the ratio of plasmids to transfection reagent was 1 μg to 2 μL with 120 μL Opti-MEM in 6-well plates.

### 2.4. RNA Extraction and RT-qPCR

Total RNAs in the cell samples were extracted using Trizol reagent. A total of 1 μg of RNA was reverse transcribed to cDNA with a StarScript II RT Mix with a gDNA Remover kit. RT-qPCR was conducted with 1μL of cDNA as the template using 2×RealStar Power SYBR qPCR Mix according to the manufacturer’s instructions. The GAPDH gene was used as the internal control. The results were calculated with the comparative CT (2^−ΔΔCT^) method. The primers used in this study are listed in Table 2.

### 2.5. Dual-Luciferase Reporter Assays

HEK-293T cells were cultured in 48-well plates and transfected with 80ng/well IFN-β-Luc expression plasmids along with 5ng/well pRL-TK, 80ng/well cGAS-HA, 80ng/well STING-HA, and indicated plasmids or control vector plasmids. At 24 h post-transfection, cells were lysed using lysis buffer diluted to the working concentration. Then the firefly and luciferase activities in the lysates were determined with the double-luciferase reporter assay kits (DL101-01, Nanjing, China) according to the manufacturer’s protocol.

### 2.6. Western Blotting

HEK-293T and MDBK cells were cultured in 6-well plates, and the cells were transfected with indicated plasmids or infected with HSV-1, HSV-GFP, or LSDV. After the indicated time of transfection, cells were collected and resolved by RIPA lysis buffer (P0013C, Beyotime, Shanghai, China) in an ice bath for 15 min. Then an appropriate amount of protein loading buffer was added and the samples were heated at 100 °C to fully denature the protein. Each protein component in the sample was then separated by electrophoresis using SDS-PAGE and transferred to solid-phase carrier PVDF membranes. The membranes were then blocked with 5% skim milk for 1 h. The excess PVDF membranes were cut off according to the position of the protein marker and incubated with the corresponding primary antibody for 8–10 h at 4 °C. Then the PVDF membranes were washed with TBST and incubated with the corresponding secondary antibody at room temperature for 1 h. Finally, ECL was used to visualize and analyze the results in a gel imager (GE Healthcare, GE-AI600, Boston, MA, USA).

### 2.7. Coimmunoprecipitation (Co-IP)

HEK-293T cells were cultured in 100 mm dishes and transfected with the indicated plasmids. After 24–36 h culture in 37 °C incubators, cells were collected by phosphate buffer and lysed by using NP-40 lysis buffer containing 1% protease inhibitor in an ice bath for more than 15 min. The supernatant was collected by centrifugation; an appropriate amount of supernatant was added to the protein loading buffer, and they were mixed well and heated at 100 °C for 10 min to prepare the input samples. The remaining samples were divided into two portions and the corresponding antibodies were added, then the samples were incubated at 4 °C for 8–12 h. The incubated samples were added to NP-40 equilibrated agarose beads and incubated at 4 °C for 4–6 h. After incubation, the agarose beads were washed with NP-40, an appropriate proportion of protein loading buffer was added, and the samples were heated at 100 °C for 10 min to complete the sample preparation. After Co-IP, Western blotting tests were performed to analyze the results.

### 2.8. Indirect Immunofluorescence Assay (IFA)

HEK-293T cells were cultured in a special confocal culture dish and transfected with the indicated plasmids. At 24 h post-transfection, the cells were fixed with 4% paraformaldehyde for 15 min. Then, the samples were treated with 0.2% Triton X-100 for 5 min and blocked with 5% bovine serum albumin (BSA) for 30 min. The corresponding antibody labeled as the primary antibody was incubated at 4 °C for 8–12h. Then the corresponding fluorescent-dye conjugated secondary antibody was incubated in a dark environment for 30 min at 37 °C and finally the nuclei were stained with DAPI for 5 min. The cell culture dish was placed under a laser confocal microscope (ZEISS, Jena, Germany) to observe the localization of intracellular proteins.

### 2.9. Establishment of a BMEC Cell Line Stably Expressing LSDV ORF137

Lentiviral packaging technology was used to construct a stably expressed ORF137 protein. Firstly, the ORF137 gene was cloned into the pLVX-mCherry-N1 with *Xho I* and *BamH I* enzyme digestion to construct the lentivirus packaging overexpression plasmids. Secondly, pMD2.G, psPAX2, and pLVX-mCherry-N1-LSDV ORF137 were co-transfected into HEK-293T cells. At 48 h post-transfection, we collected the supernatant and added it into BMEC to infect them with packaged lentivirus. Then, 24 h post-infection, BMEC were cultured with a medium containing an appropriate concentration of puromycin, and the medium was replaced every 2d ays until all cells were resistant to puromycin. Western blotting and RT-qPCR were used to verify the expression of ORF137 in BMEC.

### 2.10. LSDVΔORF137-EGFP Strain Construction

The recombinant LSDVΔORF137-EGFP strain was generated by homologous recombination method using the parental LSDV genome and a recombination transfer vector. Firstly, the upstream and downstream sequences of the ORF137 (462 bp) gene were designed as homologous recombination arms and cloned into the backbone vector *p*UC-19T-EGFP. The nucleotide sequence of the left recombination arm is identical to the nucleotides 128,798–129,444 (137 L) of LSDV/CHA/FJ/2021, and the nucleotide sequence of the right recombination arm is identical to the nucleotides 129,941–130,205 (137 R) of LSDV/CHA/FJ/2021. Next, the 137L-EGFP-SV40 polyA-137R was inserted into the recombinant transfer vector to construct the plasmid *p*UC19T-LSDVΔORF137-EGFP. Therefore, the transfer plasmid *p*UC19T-LSDVΔORF137-EGFP was transfected into Vero cells, followed by infection of the cells with LSDV/CHA/FJ/2021. After 24 h, the recombinant virus infection was observed under a fluorescence microscope (Invitrogen EVOS, Carlsbad, USA). Finally, the purified LSDVΔORF137-EGFP was obtained through limited dilution, amplification culture, purity detection, and PCR determination of the target gene. The deleted ORF137 is located at the site of 129,444–129,941 of the full genome sequence of LSDV/CHA/FJ/2021. The primers used in this study are listed in Table 3.

### 2.11. Viral Growth Kinetics

To compare the differences between the replication of WT-LSDV and LSDVΔORF137-EGFP, the growth kinetics of the two viruses were examined. A six-well plate was inoculated with about 1 × 10^5^ MDBK cells. After incubation overnight, the cells were infected with WT-LSDV (10^7.515^TCID_50_) or LSDVΔORF137-EGFP (10^5.55^TCID_50_) at an MOI of 1. After incubation for 12, 24, 48, 72, and 96 h, the supernatant was collected and the viral titer was measured as the median tissue culture infective dose (TCID_50_).

### 2.12. Statistical Analysis

The data in this study were analyzed by GraphPad Prism (Insightful Science, San Diego, USA) (Version 8.01) software. Part of the data were analyzed by one-way analysis of variance (ANOVA) followed by Dunnett’s post hoc test to compare one treated group with the non-treated group and Tukey’s post hoc test for multiple comparisons between groups. The other part of the data was analyzed by Student’s t-test. The methods above were used for comparison with at least three independent trials (ns: non-significant, * *p* < 0.05, ** *p* < 0.01, *** *p* < 0.001, and **** *p* < 0.0001).

## 3. Results

### 3.1. LSDV ORF137 Inhibits cGAS-STING-Induced Transcriptional Activation of IFN-β

In order to screen the viral proteins encoded by LSDV-inhibiting cGAS-STING-induced IFN-β promotor activation, HEK-293T cells were transfected with 156 eukaryotic expression plasmids expressing LSDV-encoding proteins and IFN-β luciferase reporter plasmids to perform dual-luciferase reporter assay. These assays identified several LSDV proteins that had significant effects on reporter gene expression. Of these, the LSDV ORF137 protein significantly inhibited the activation of IFN-β induced by cGAS-STING to the greatest extent and was selected for further analyses in this study (Figure 1A). The results for the remaining LSDV proteins will be reported elsewhere. To reconfirm whether the LSDV ORF137 protein inhibits the activation of IFN-β through the cGAS-STING pathway, HEK-293T cells were treated with or without the transfection of cGAS and STING, and different doses of the LSDV ORF137 expression plasmid together with the IFN-β reporter plasmid were co-transfected into the cells, followed by luciferase assays. The results indicate that, in experiments with the transfection of cGAS and STING plasmids, the overexpression of LSDV ORF137 showed a dose-dependent inhibitory effect on cGAS-STING-mediated IFN-β activation during HSV-1 infection (Figure 1B). In contrast, in experiments without the addition of cGAS and STING plasmids, the overexpression of ORF137 slightly affected IFN-β promoter activation-mediated HSV-1 infection, and there was no dose dependence (Figure 1C). These results indicate that LSDV ORF137 negatively regulates the host’s innate immune response by targeting the cGAS-STING pathway.

### 3.2. LSDV ORF137 Protein Inhibits the Antiviral Innate Immune Response

Interferon-stimulated genes (ISGs) and IFN-induced GTP-binding protein Mx-1 (Mx-1) are a class of antiviral factors induced by IFN which play a crucial role in host resistance to viral infections [23]. To explore the specific inhibition of ORF137 protein on the production of IFN-β and antiviral factors mediated by cGAS-STING at the mRNA level, ORF137 or the control vector pCAGGS-Flag was co-transfected with cGAS and STING into HEK-293T cells. After 6 h, the cells were infected with HSV-GFP, and 24 h later, the samples were lysed with Trizol to extract the mRNA. Quantitative reverse transcription-PCR (RT-qPCR) was used to analyze the mRNA levels of IFN-β and IFN-β-induced ISG54, ISG56, and Mx-1. The results indicate that LSDV ORF137 inhibits the transcription of IFN-β, ISG54, ISG56, and Mx-1 genes induced by cGAS-STING in HEK-293T cells post-HSV-GFP-infection (Figure 2A–D). Meanwhile, both fluorescence microscopy imaging and Western blotting analysis results show that the LSDV ORF137 protein can promote the replication of HSV-GFP mediated by the cGAS-STING signaling pathway (Figure 2E,F). In conclusion, these results indicate that LSDV ORF137 protein inhibits the antiviral response mediated by the cGAS-STING pathway, thereby promoting viral replication.

### 3.3. ORF137 Is Transcribed in the Early Stages of Infection and Localizes in the Cytoplasm

In order to explore the transcriptional kinetics of the ORF137 gene during the viral life cycle, MDBK cells were infected with 1 MOI of LSDV, and cell samples were collected at 0, 6, 12, and 24 h post-infection. As shown in Figure 3A,B, the expression of ORF137 mRNA was detected by RT-qPCR as early as 6h post-infection (Figure 3A), and the expression of the ORF137 protein can also be clearly detected by Western blotting as early as 6 h post-infection (Figure 3B). To determine the intracellular localization of LSDV ORF137 protein in infected cells, ORF137-Flag and the control vector pCAGGS-Flag were separately transfected into HEK-293T cells for 24 h; similarly, MDBK cells were infected with LSDV at 1 MOI for 24 h. Subsequently, an IFA experiment was conducted. As shown in Figure 3C,D, compared to the control vector pCAGGS-Flag, bright green fluorescence was observed in HEK-293T cells overexpressing ORF137, which was localized in the cytoplasm (Figure 3C). Furthermore, compared to uninfected cells, bright red fluorescence was observed in the cytoplasm of LSDV-infected MDBK cells by using a ORF137 antibody, showing a speckled pattern, clearly indicating that the ORF137 protein is located in the cytoplasm (Figure 3D).

### 3.4. LSDV ORF137 Interacts with Homo Sapiens-IRF3 (Hs-IRF3)/Bos Taurus-IRF3 (B-IRF3)

To investigate the molecular mechanisms of ORF137 in regulating the type-I interferon immune response, we first determined whether ORF137 is associated with the components involved in DNA virus-triggered signaling pathways. First, we predicted the potential protein interactions between the key molecules cGAS, STING, TBK1, and IRF3 of the cGAS-STING signaling pathway and ORF137 using AlphaFold 3 (https://alphafoldserver.com (Accessed on: 9 July 2024)); the results indicated that AlphaFold 3 predicted the highest confidence of interaction between ORF137 and IRF3 (Figure 4A). To further confirm the above prediction, the plasmids of key node molecules including hs-cGAS-HA, hs-STING-HA, hs-TBK1-HA/Bos-TBK1-HA, and hs-IRF3-HA/Bos-IRF3-HA were co-transfected with ORF137-Flag into HEK-293T cells. Additionally, Bos-cGAS-Flag and Bos-STING-Flag were co-transfected with ORF137-HA into HEK-293T cells. The cell lysates were immunoprecipitated using HA Rabbit or Flag Mouse mAb, and the co-immunoprecipitated proteins were analyzed by Western blotting using the corresponding antibodies (Figure 4B,C and Appendix A). The reciprocal immunoprecipitation experiments showed that ORF137 interacted with hs-IRF3-HA/Bos-IRF3-HA (Figure 4B,C), whereas ORF137 did not interact with cGAS, STING, or TBK1 in our experiments (Appendix A). Furthermore, the co-localization of ORF137 and IRF3 was conducted using confocal microscopy through IFA in HEK-293T cells. The results showed an obvious co-localization of ORF137 and IRF3 in the cytoplasm (Figure 4D). Collectively, the data indicated that the LSDV ORF137 specifically interacted with IRF3.

### 3.5. Overexpression of LSDV ORF137 in BMEC Promotes LSDV Replication Through Inhibiting the Transcription of IFN-β and ISGs

To further verify the capacity of LSDV ORF137 to modulate innate immune signaling, we examined the effect of ORF137 on the activity of a downstream signaling molecule, IRF3. Transfection of cGAS-HA, STING-myc, ORF137-Flag, and the control vector pCAGGS-Flag plasmids into HEK-293T cells occurred over 24 h, then the cells were infected with HSV-1. Cells were collected and we detected the expression and phosphorylation of IRF3 using Western blotting. The results indicate that, compared with the cells transfected with the control plasmid, the overexpression of ORF137 did not affect the expression of IRF3, but decreased the phosphorylation of IRF3 during HSV-1 infection in HEK-293T cells (Figure 5A). Furthermore, lentiviral packaging technology was used to establish cell lines stably expressing ORF137 protein in BMEC (Figure 5B). As shown in Figure 5C,D, compared with control cells, the cell line showed stable expression of ORF137 at the transcriptional and protein level (Figure 5C,D). Subsequently, the replication of LSDV in different cells was detected through Western blotting and RT-qPCR assays. As shown in Figure 5E, compared with the control cells, the cells overexpressing ORF137 showed a decreased phosphorylation of IRF3 and increased replication of LSDV. In addition, the transcription of the IFN-β, ISG54, ISG56, and Mx-1 genes was significantly suppressed in ORF137-SE cells compared with the control (Figure 5F). Overall, these results showed that the overexpression of ORF137 can promote the replication of LSDV by antagonizing the phosphorylation of IRF3, thereby inhibiting the transcriptional level of IFN-β and ISGs.

### 3.6. Construction and Biological Evaluation of ORF137 Deletion LSDV Strain

To further study the function of ORF137 during LSDV infection, ORF137 was deleted from the wild-type LSDV/FJ/CHA/2021 (GenBank: 37336054) strain through homologous recombination based on the schematic diagram (Figure 6A). Through PCR amplification, limited dilution of positive cells, and Western blotting verification, we successfully constructed the ORF137 deletion LSDV strain LSDVΔORF137-EGFP (Figure 6B–E). Firstly, MDBK cells were infected with the wild-type LSDV (WT-LSDV) or LSDVΔORF137-EGFP strain, the virus titer in MDBK cells at different time points was determined, and their growth curves were plotted. The results showed that, compared with the WT-LSDV strain, the deletion of ORF137 reduced the replication of mutant virus in MDBK cells at 24 h post-infection (hpi). At 48 hpi, the replication of LSDVΔORF137-EGFP showed no obvious difference compared with that of LSDV (Figure 6F). The RT-qPCR results showed that, compared with control, the deletion of ORF137 significantly increased the transcription of IFN-β, ISG54, ISG56, and Mx1 (Figure 6G). To explore the underlying mechanism, we further investigated the effect of viral infection on the interferon signaling pathway. As shown in Figure 6H, we found that, compared with the WT-LSDV, infection by the LSDVΔORF137-EGFP strain showed no significant effect on IRF3 expression but had higher phosphorylation of IRF3 at 6 h after virus infection, which corresponds to lower virus replication at 6 h; however, with the increase in time, the difference in IRF3 phosphorylation on virus replication was not obvious, indicating enhanced activation of interferon signaling during the early stage of virus infection. Based on the above results, ORF137 may play a role in the inhibition of interferon activation and ISG transcription during early infection of LSDV.

## 4. Discussion

Most viral infections trigger a series of signal cascades that lead to the expression of type I IFN and hundreds of interferon-stimulating genes (ISG), which play a key role in the cellular antiviral response [24,25]. It has been reported that the cGAS-STING DNA sensing pathway induces IFN production and inhibits the replication of viruses, especially DNA viruses [26,27]. A large amount of experimental data shows that ASFV, which is also a large double-stranded DNA virus, encodes many proteins, including pI215L [28], I226R [29], A137R [30], pD345L [31], DP96R [32], MGF360-14L [33], MGF505-11R, UBCv1 [34], S273R [35], MGF360-11L [36], and MGF-505-7R [37], all of which have different mechanisms to inhibit the cGAS-STING signaling pathway to resist host immune defenses. LSDV encodes more than 150 proteins, some of which may establish complex interactions with the host to benefit the viral invasion and evade the host’s defenses [2]. Recent reports have indicated that among the proteins encoded by the lumpy skin disease virus (LSDV), ORF 001/156 acts as a negative regulator of the interferon (IFN) signaling pathway. It interacts with interferon regulatory factor 3 (IRF3), disrupting the latter’s dimerization and nuclear translocation, thereby impairing IFN production [19]; ORF 012 interacts with the IFN-induced protein IFIT1 and alters its subcellular localization. Further studies have demonstrated that the deletion of ORF 012 from LSDV significantly affects viral replication, highlighting the importance of LSDV012 in antagonizing the type I IFN response [17]. The ORF 103 protein can induce a strong humoral and cellular immune response, characterized by a significant increase in the levels of IgG, IFN-γ, IL-1β, and CD3^+^CD4^+^T cells. This suggests that ORF 103 is a highly immunogenic protein with strong diagnostic specificity [38]. ORF 127 exerts a negative regulatory effect on the expression of IFN-β via the cGAS-STING signaling pathway [16]. ORF 132 is a key gene for LSDV replication, playing a crucial role in the viral life cycle, and it also functions as a negative regulator of endoplasmic reticulum stress [39]. ORF 142 antagonizes the host’s antiviral innate immunity by interfering with the binding between TANK-binding kinase 1 (TBK1) and IRF3 [40]. Additionally, other studies have found that ORF 142 inhibits the cGAS/STING-mediated type I IFN pathway through NBR1-mediated autophagic degradation of STING [41]. Furthermore, in studies on the pathogenesis of LSDV, the presence of the viral genome, viral replication (in blood, skin nodules, and lymph nodes), and strain-specific virulence have been confirmed. This also poses considerable challenges for our efforts to elucidate the functions of LSDV proteins [42]. However, sufficient experimental evidence is still lacking regarding the role of LSDV proteins in regulating host immune response. In this study, our results indicate that LSDV ORF137 negatively regulates type I IFN through IRF3 signaling node molecules mediated by the cGAS-STING pathway.

The protein sequence of ORF137 (LSDV/FJ/CHA/2021, GenBank: OP752701.1) was analyzed by Blast in the National Center for Biotechnology Information (NCBI) of the United States (https://blast.ncbi.nlm.nih.gov/Blast.cgi (Accessed on: 1 July 2025)), and we found that the function of ORF137 was predicted to be similar to that of the A52R-like protein family, which is found in many poxviruses such as Goatpox virus (GTPV, GenBank:AGZ95451.1), Swinepox virus (SWPV, GenBank: NP570293.1), Hypsugopox virus (HYPV, GenBank: QDJ95119.1), Eptesipox virus (EPTV, Taxonomy ID: 1329402), Cowpox virus (CPXV, GenBank: UPV00425.1), and Vaccinia virus (VV, Gene ID: 3707707). Moreover, a number of studies about Vaccinia virus (VV) have shown that the A52R protein participates in regulating host immune response [43,44,45,46]. In addition, LSDV ORF137 is presumed to belong to the Poxvirus Bcl-2-like family (https://www.ebi.ac.uk/interpro/entry/InterPro/IPR022819/ (Accessed on: 1 July 2025)), which includes proteins B14 [47], A52 [48], A46 [49], C6 [50], K7 [51], N1 [52], and N2 [53]. These proteins have low sequence identity but exhibit high structural similarity to the eukaryotic Bcl-2 protein family [54,55]. Most of them share a Bcl-2 fold, which consists of a central hydrophobic α helix surrounded by an additional layer of 6–7 amphipathic α helices [48]. Studies have shown that poxvirus Bcl-2-like proteins function as immune modulators, evading host innate immune responses such as interferon (IFN) regulatory factor-3 (IRF-3) and nuclear factor-kappaB (NF-κB) by inhibiting apoptosis or blocking the activation of pro-inflammatory transcription factors [56]. However, the above viral proteins with high sequence homology or structural similarity can only provide a reference for predicting the function of LSDV ORF137 proteins, and experiments are still needed to determine the function of ORF137 proteins. In this study, we used a dual-luciferase reporter system to screen the effects of LSDV-encoded proteins on IFN- β promoter activation and found that LSDV ORF137 is one of the strongest inhibitors of cGAS-STING-induced IFN-β promoter activation (Figure 1). More interestingly, ORF137 overexpression significantly inhibited the transcriptional expression of IFN-β, ISG54, ISG56, and Mx1 in HEK-293T cells (Figure 2). To explore the function of ORF137 in host cells, we constructed BMEC cell lines stably expressing ORF137 using lentiviral packaging technology. Further studies showed that overexpression of ORF137 in BMEC could promote LSDV replication by inhibiting the expression of IFN-β and some ISGs (Figure 5). In addition, the mRNA and protein expression of ORF137 can be detected as early as 6 h after LSDV infection (Figure 3). At the initial stage of the LSDV infection, the expression of IFN-β and its related ISGs decreased first and then increased (Figure 6G). This has been similarly shown in previous studies [16], which implies an inhibitory function of some LSDV components during early viral infection. However, the absence of ORF137 weakened this inhibitory effect, suggesting that the LSDV ORF137 gene may be one of the key proteins in inhibiting the production of IFN-β and promoting viral replication in the early stage of viral infection.

IRF 3 is a key transcription factor shared by various PRRs and is important in the induction of IFN-I and essential for viruses eliciting the expression of many host genes involved in the innate immune response [57]. There is a large amount of evidence that viruses have evolved multiple strategies to inhibit IRF3 activation and evade the host’s antiviral response. The EP364R [58], pI215L [59], and A137R [30] proteins of African swine fever virus (ASFV) can block the activation of the cGAS-STING pathway and inhibit the activity of IRF3 and the production of IFN-I. The NS4B of Zika virus (ZIKV) binds to DHCR7 to inhibit TBK1 and IRF3 activation, which in turn inhibits IFN-β and ISGs, thus promoting ZIKV immune evasion [60]. The nsp15 viral protein of both Porcine Epidemic Diarrhea Virus (PEDV) and severe acute respiratory syndrome coronavirus 2 (SARS-CoV-2) can antagonize interferon signaling by degrading or inhibiting IRF3 [27,61]. The VP1 of foot-and-mouth disease virus (FMDV) degrades YTHDF2 through autophagy to regulate IRF3 activity, and it also promotes viral replication by antagonizing TPL2-mediated activation of the IRF3/IFN-β signaling pathway [62]. In addition, the nuclear localization signal of monkeypox virus (MPXV) protein P2 orthologue is critical for inhibition of IRF3-mediated innate immunity [63]. The viral proteins N2 [53], C6 [64], and A46 [49] of vaccinia virus (VACV) can affect the activity of IRF3 through different pathways and contribute to the virulence of VACV. It should be emphasized that, like LSDV, monkeypox virus (MPXV) and vaccinia virus (VACV) are large DNA poxviruses that encode a large number of proteins. In this study, our data suggest that the LSDV ORF137 protein interacts with IRF3 (Figure 4). Overexpression of the ORF137 protein decreases the phosphorylation of IRF3 (Figure 5A, Figure 5E), which in turn inhibits the expression of IFN-β and ISGs (Figure 5F, Figure 6G), and ultimately facilitates LSDV replication (Figure 5D, Figure 6H).

## 5. Conslusions

Based on our results, we propose a schematic model of the innate antiviral response mediated by LSDV ORF137 (Figure 7). During LSDV infection, LSDV ORF137 interacts with IRF3 and antagonizes the phosphorylation of IRF3, thereby inhibiting the production of interferons and expression of interferon-stimulating genes. In the absence of ORF137, the suppression of IRF3 phosphorylation during the early LSDV infection was attenuated, which led to high expression IFN-β and some ISGs. This study may provide new insights into the immune evasion mechanisms involved in viral infection and the development of vaccination strategies to combat global LSDV spread.

## Figures and Tables

**Figure 1 cells-14-01475-f001:**
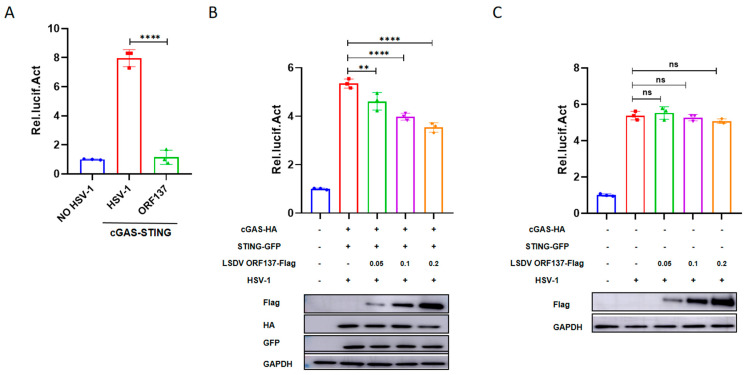
LSDV ORF137 Inhibits IFN-β activity dependent on cGAS-STING pathway. (**A**) HEK-293T cells were cultured in 48-well plates and co-transfected with pRL-TK (5 ng), IFN-β-Luc (80 ng), cGAS (100 ng), STING (100 ng), pCAGGS-Flag (100 ng), and different LSDV eukaryotic expression plasmids (100 ng) for 12 h, stimulated with HSV-1 (HSV) for 12 h, and then the cell pellets were collected to detect luciferase activity. (**B**) HEK-293T cells were cultured in 24-well plates and co-transfected with pRL-TK (10 ng), IFN-β-Luc (160 ng), cGAS (200 ng), STING (200 ng), pCAGGS-Flag (200 ng), and pCAGGS-ORF137-Flag (50, 100, and 200 ng) for 12 h, then stimulated with HSV-1 for 12 h, and the cell pellets were collected to detect luciferase activity. The collected cell pellets were also subjected to Western blotting experiments to quantify GAPDH, STING, cGAS, and ORF137 protein expression. (**C**) HEK-293T cells were cultured in 24-well plates and co-transfected with pRL-TK (10 ng), IFN- β -Luc (160 ng), pCAGGS-Flag (200 ng), and pCAGGS-ORF137-Flag (50, 100, and 200 ng) for 12 h, followed by HSV-1 stimulation for 12 h, then we collected the cell pellets to detect luciferase activity. (Data are shown as the SD ± mean value. Analysis of variance and t-tests were performed for statistical analyses; ns: non-significant, * *p* < 0.05, ** *p* < 0.01, *** *p* < 0.001, and **** *p* < 0.0001).

**Figure 2 cells-14-01475-f002:**
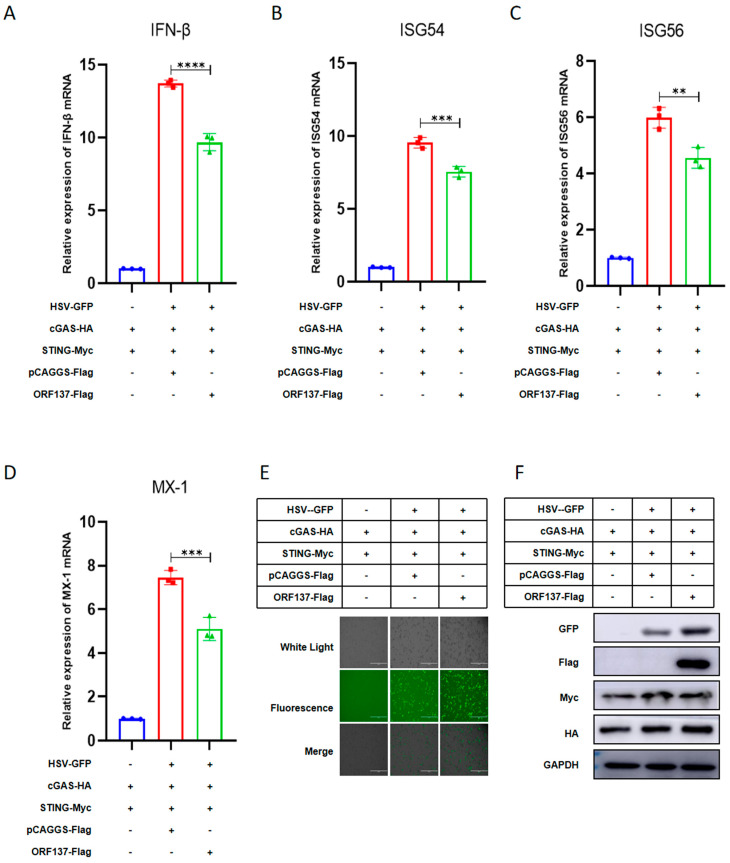
LSDV ORF137 inhibits antiviral innate immune response. (**A**–**D**) HEK-293T cells were co-transfected with plasmids cGAS-HA (0.8 μg) and STING-Myc (0.8 μg) and control vector pCAGGS-Flag (1 μg) or ORF137-Flag (1 μg) in 6-well plates for 24 h, followed by HSV-GFP stimulation for 12 h, and samples were collected with Trizol reagent. RT-qPCR analysis was performed to quantify the transcription of IFN-β, ISG54, ISG56, and Mx-1. (**E**,**F**) In parallel, fluorescence microscopy was employed to observe and photograph viral replication of HSV-GFP, and samples treated with cell lysate buffer under the same treatment conditions as above underwent Western blotting experiments to assess the proteins’ expression of GAPDH, STING, cGAS, ORF137, and HSV. (Data are shown as the SD ± mean value, and t-tests were performed for the statistical analyses; ns: non-significant, * *p* < 0.05, ** *p* < 0.01, *** *p* < 0.001, and **** *p* < 0.0001).

**Figure 3 cells-14-01475-f003:**
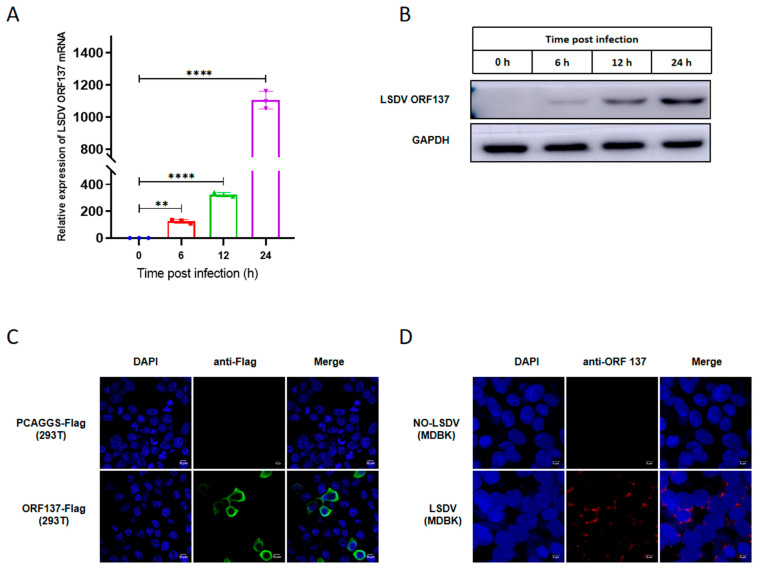
The expression and intracellular localization of LSDV ORF137. (**A**) MDBK cells were infected with LSDV at 1 MOI for 0, 6, 12, and 24 h. Samples were collected, and the transcriptional level of LSDV ORF137 was detected by RT-qPCR. (**B**) MDBK cells were infected with LSDV at MOI 1, cells were collected at indicated times, and Western blotting was performed to determine the expression of ORF137; GAPDH was used as an internal control to show the even loading of samples. (**C**) HEK-293T cells were transfected with ORF137-Flag for 24 h. Localization of ORF137 proteins in the cells was visualized by IFA using anti-FLAG antibodies. (**D**) MDBK cells were infected with LSDV at 1 MOI for 24 h. Localization of ORF137 proteins in the cells was visualized by IFA using ORF137 polyclonal antibodies. (Data are shown as mean ± SD of three independent experiments and an ANOVA was performed for statistical analysis; ns: non-significant, * *p* < 0.05, ** *p* < 0.01, *** *p* < 0.001, and **** *p* < 0.0001).

**Figure 4 cells-14-01475-f004:**
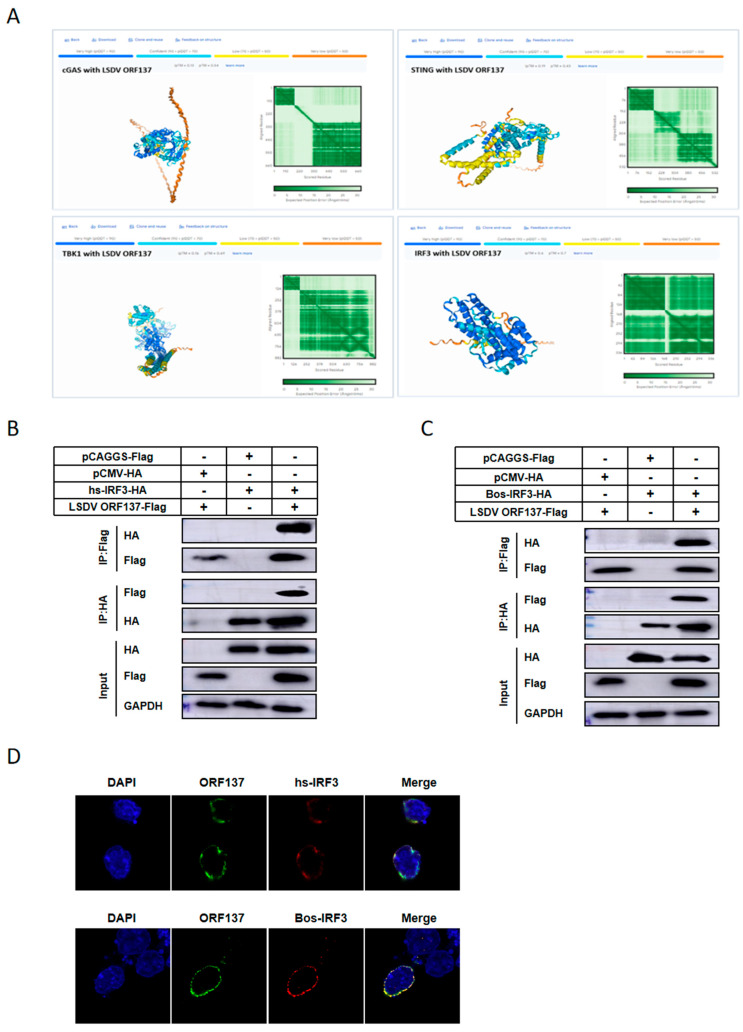
LSDV ORF137 interacts with homo sapiens-IRF3 (hs-IRF3)/Bos taurus-IRF 3 (B-IRF3). (**A**) Use AlphaFold3 Server online to predict the interaction possibility pattern diagram of LSDV ORF137 protein with key proteins regulating innate immune cGAS-STING signaling pathway, namely cGAS, STING, TBK-1, and IRF3 proteins. Different colors correspond to different confidence levels, with the confidence decreasing in the order of dark blue, light blue, yellow, and orange. For prediction errors, the greener the color, the smaller the error; if the protein as a whole is dark blue, it indicates the good confidence. (**B**,**C**) hs-IRF3-HA (4 μg) and Bos-IRF3-HA (4 μg) were, respectively, co-transfected with control vector pCAGGS-Flag (4 μg) or ORF137-Flag (4 μg) into HEK-293T cells for 24 h. Cell lysates were immunoprecipitated with HA or Flag antibodies, and the co-precipitated proteins were analyzed by Western blotting with the indicated antibodies. (**D**) HEK-293T cells were transfected with ORF137-Flag (1 μg) and hs-IRF3-HA (1 μg) or Bos-IRF 3-HA (1 μg) plasmids; 24 h later, the localization of ORF137 with IRF3 was observed by IFA.

**Figure 5 cells-14-01475-f005:**
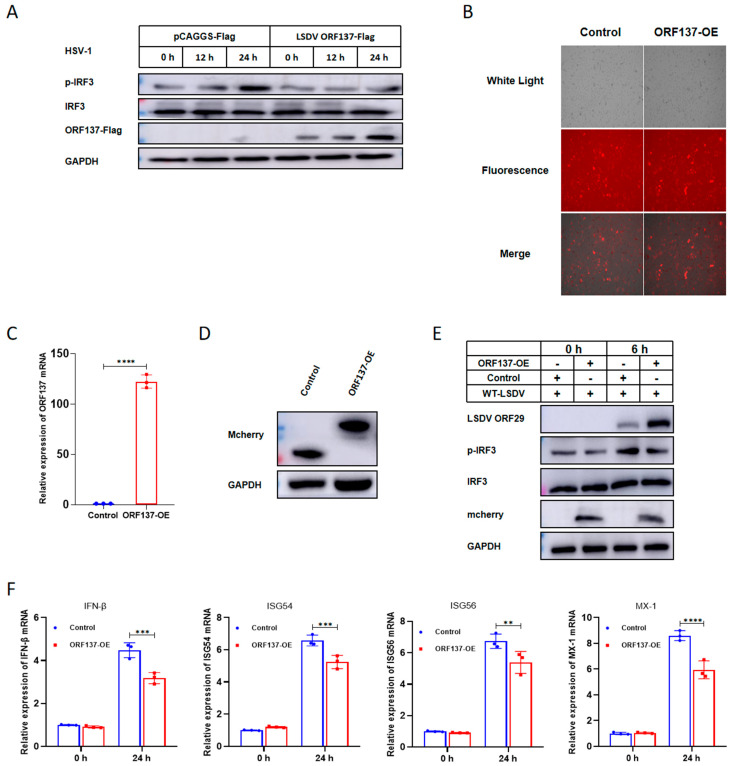
Overexpression of LSDV ORF137 in BMEC promotes LSDV replication through inhibiting the transcription of IFN-β and ISGs. (**A**) HEK-293T cells were co-transfected with cGAS-HA (0.8 μg), STING-Myc (0.8 μg), and the control vector pCAGGS-Flag (1 μg) or ORF137-Flag (1 μg) in a 35 mm cell culture dish for 12 h, followed by HSV-1 at 1 MOI for 0, 12, and 24 h. Western blotting assay was performed with the indicated antibodies. (**B**) The plasmids pMD2.G, psPAX2, and pLVX-mCherry-N1-LSDV ORF137 were co-transfected into HEK-293T cells at a ratio of 1:2:3. At 48 h post-transfection, we collected the supernatant and added it into BMEC to infect them with packaged lentivirus. Then, 24 h post-infection, BMEC were cultured with a medium containing an appropriate concentration of puromycin, and the medium was replaced every 2days until all cells were resistant to puromycin. The successfully constructed ORF137 protein overexpression cells (ORF137-OE cells) and control cells, both carrying the red fluorescent label mCherry, were established in BMEC. (**C**,**D**) RT-qPCR and Western blotting were used to determine the expression of ORF137 at the transcriptional and protein levels with the specified primers and antibodies. (**E**) Control and ORF137-OE cells were infected with LSDV at 1 MOI, and cells were collected at 0 and 24 h after LSDV infection. Western blotting was employed to detect the protein expression of LSDV ORF137, IRF3, Phospho-IRF3 and LSDV ORF29. (**F**) Control and ORF137-OE cells were infected with LSDV at 1 MOI. Samples were collected at 0 and 24 h after LSDV infection and RNA were extracted by Trizol reagent; the transcriptional levels of IFN- β, ISG54, ISG56, and Mx 1 were detected by RT-qPCR. (Data are shown as mean ± SD of three independent experiments and we performed an ANOVA for our statistical analysis; ns: non-significant, * *p* < 0.05, ** *p* < 0.01, *** *p* < 0.001, and **** *p* < 0.0001).

**Figure 6 cells-14-01475-f006:**
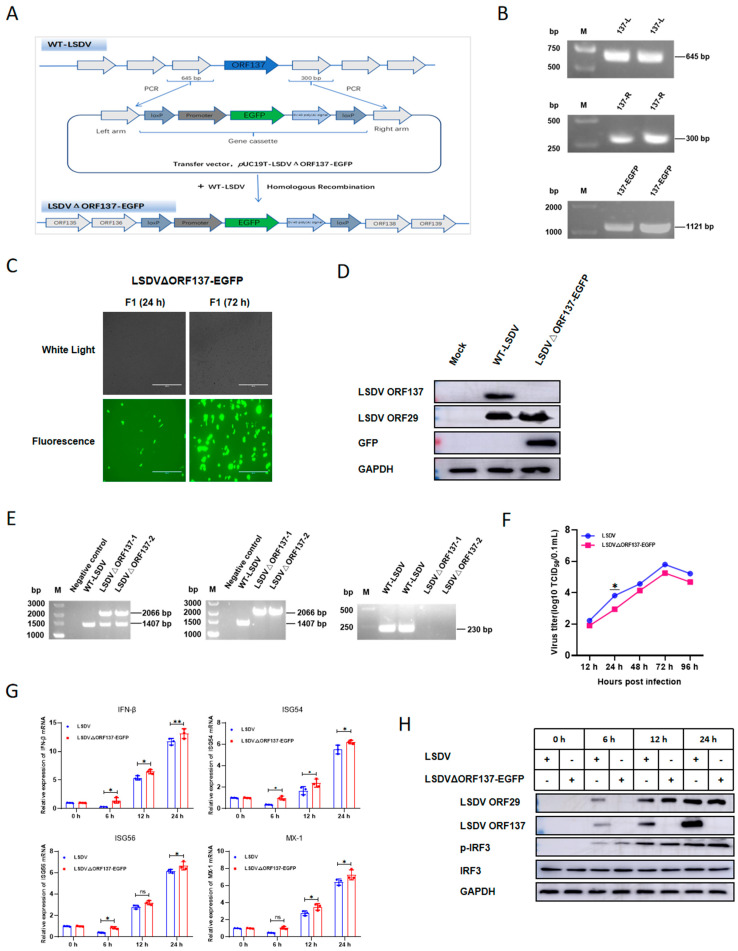
Construction and biological evaluation of ORF137 deletion LSDV strain. (**A**) Schematic diagram for constructing the ORF137 deletion recombinant LSDV strain. (**B**) PCR amplification of different fragments for plasmid construction. From top to bottom, they are indicated as follows: the left homologous arm amplified by primer pair 1 (645 bp), the right homologous arm amplified by primer pair 2 (300 bp), and the gene cassette amplified by primer pair 3 (1121 bp). The gene cassette contains two loxP sites, the EGFP reporter gene and pmH5 poxvirus promoter. (**C**) The fluorescence observation of LSDVΔORF137-EGFP under microscope. Vero cells were transfected with pUC19T-LSDVΔORF137-EGFP plasmid, then infected with LSDV 24 h and 72 hpi, the ORF137 deletion LSDV was observed through green fluorescence. (**D**) Verification of the ORF137 deletion strain LSDVΔORF137-EGFP by Western blotting. Cells were infected with LSDV (MOI = 1) or LSDVΔORF137-EGFP (MOI = 1); 48 h later, samples were collected and Western blotting was performed to detect protein expression with the indicated antibodies. (**E**) Verification of the ORF137 deletion strain LSDVΔORF137-EGFP by PCR. From left to right, the diagrams are indicated as follows: the left and middle PCR result diagrams are amplified using the primer pair Primer1-137-Forward and Primer2-137-Reverse. Compared with the WT-LSDV, the purification of the recombinant virus will be successful only when a single band appears in the LSDVΔORF137-EGFP; the right panel of the PCR result diagram is amplified using primer pair 4. Compared with the WT-LSDV, the purification of the recombinant virus will be successful only when no band appears in the LSDVΔORF137-EGFP. (**F**) Growth kinetics curves of LSDV and LSDVΔORF137-EGFP in the MDBK cells. MDBK cells were infected with WT-LSDV or LSDVΔORF137-EGFP at 1 MOI for the indicated times and samples were collected for TCID_50_ assay following the protocol in the Materials and Methods. (**G**) MDBK cells were infected with LSDV or LSDVΔORF137-EGFP at 1 MOI and harvested at specific time points after infection (0, 6, 12, and 24 h) and then analyzed for mRNA expression of IFN-β, ISG54, ISG56, and Mx-1 using RT-qPCR. (**H**) MDBK cells were infected with LSDV or LSDVΔORF137-EGFP at 1 MOI, and Western blotting analysis was performed to detect the expression of IRF3, p-IRF3, ORF137, ORF29, and GAPDH using the indicated antibodies. (Data are shown as mean ± SD of three independent experiments and we performed an ANOVA for our statistical analyses; ns: non-significant, * *p* < 0.05, ** *p* < 0.01, *** *p* < 0.001, and **** *p* < 0.0001).

**Figure 7 cells-14-01475-f007:**
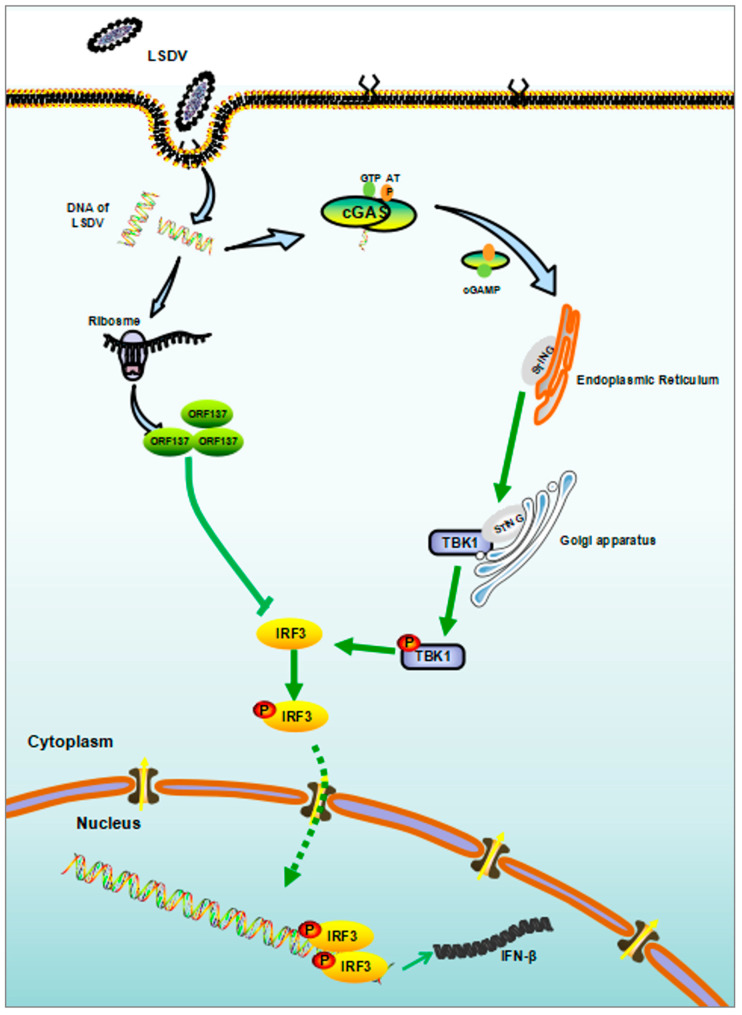
A schematic model of LSDV ORF137 function during viral infection. During LSDV infection, cells initiate the cGAS-STING signaling pathway to combat the virus; to evade the host immune response, LSDV expresses the ORF137 protein to interact with IRF3 and decrease its phosphorylation, which leads to reduced production of interferons and ISGs, eventually facilitating the replication of LSDV during early infection.

**Table 1 cells-14-01475-t001:** The source and application of the antibodies in this study.

Reagent Name	City/Country	Company Identification	Ratio	Item Number
GAPDH Polyclonal antibody	Wuhan/China	Proteintech	1: 5000	10494-1-AP
DYKDDDDK Tag (FLAG) Recombinant Rabbit Monoclonal Antibody [PSH07-02]	Hangzhou/China	huabio	1: 3000	HA722780
DYKDDDDK Tag (FLAG) Recombinant Mouse Monoclonal Antibody [A2-A4-R]			1: 3000	HA601167
HA tag Recombinant Rabbit Multiclonal Antibody [PSH01-92]			1: 3000	HA721750
mCherry Mouse Monoclonal Antibody			1: 2000	HA601186
HA-Tag(26D11) mAb	Shanghai/China	Abmart	1: 2000	M20003M
IRF-3 (D83B9) Rabbit mAb	Danvers/USA	Cell signaling technology	1: 2000	#4302
Phospho-IRF-3 (Ser396) (4D4G) Rabbit mAb			1: 1000	#4947
ORF029 Mouse mAb	preserved in our laboratory			
ORF137 Mouse polyclonal antibody (pAb)				
Goat Anti-Rabbit IgG H&L (HRP)	Waltham/USA	Thermo Fisher Scientific	1: 5000	31460
Goat Anti-Mouse IgG H&L (HRP)			1: 5000	31430
Goat anti-Rabbit IgG (H+L) Highly Secondary Antibody, Alexa Fluor™ 488			1: 1000	A-11034
Goat anti-Mouse IgG (H+L) Cross-Adsorbed Secondary Antibody, Alexa Fluor™ 594			1: 1000	A-11005
4′, 6-diamidino-2-phenylindole (DAPI) solution, 1 mg/ml	Beijing/China	Solarbio	1: 1000	C0060-1 ml

**Table 2 cells-14-01475-t002:** The primer sequences used in this study.

Primers	Sequence (5′→3′)
Human-IFN-β-F (RT-qPCR)	TCTTTCCATGAGCTACAACTTGCT
Human-IFN-β-R (RT-qPCR)	GCAGTATTCAAGCCTCCCATTC
Human-ISG54-F (RT-qPCR)	ACGGTATGCTTGGAACGATTG
Human-ISG54-R (RT-qPCR)	AACCCAGAGTGTGGCTGATG
Human-ISG56-F (RT-qPCR)	CCTCCTTGGGTTCGTCTACA
Human-ISG56-R (RT-qPCR)	GGCTGATATCTGGGTGCCTA
Human-Mx1-F (RT-qPCR)	CAGGACATTTGAGACAATCGTG
Human-Mx1-R (RT-qPCR)	TCGAAACATCTGTGAAAGCAAG
Human-GAPDH-F (RT-qPCR)	GAGTCAACGGATTTGGTCGT
Human-GAPDH-R (RT-qPCR)	GACAAGCTTCCCGTTCTCAG
Bovine-IFN-β-F (RT-qPCR)	TCCTGGGGCAGTTACCTTCA
Bovine-IFN-β-R (RT-qPCR)	GAATGCCGAAGATGTGCTGG
Bovine-ISG54-F (RT-qPCR)	TCTTTCTGCCTTCTGCCTCG
Bovine-ISG54-R (RT-qPCR)	CCTTCAGTCAATGGGACGCT
Bovine-ISG56-F (RT-qPCR)	GCAGGTGACCACAGAAAAGC
Bovine-ISG56-R (RT-qPCR)	AGAAATCGGCCGTAGTGCAA
Bovine-Mx1-F (RT-qPCR)	AGTCCTCCGACTCTTCACTCA
Bovine-Mx1-R (RT-qPCR)	GTCTGCTACCAGGCCATCAA
Bovine-GAPDH-F(RT-qPCR)	TGGTGAAGGTCGGAGTGAAC
Bovine-GAPDH-R (RT-qPCR)	ATGGCGACGATGTCCACTTT
ORF137-F (RT-qPCR)	GGAGACTTGGATTTATTGTTTACTG
ORF137-R (RT-qPCR)	CCCAGCAATCCTAATAACACT

**Table 3 cells-14-01475-t003:** The primer sequences for the construction and verification of recombinant LSDV.

Name	Sequence (5′-3′)	Length
Primer1-137-Forward	ccggaattctaaaaattgttataagttaaacggttt	645 bp
Primer1-137-Reverse	cggggtaccactgtctggctctttaattataac	645 bp
Primer2-137-Forward	tgctctagatagtttttttatgtatttttatgttaaaaaataaa	300 bp
Primer2-137-Reverse	cccaagcttctctaattaaatctaccttttttttttct	300 bp
Primer3-137-EGFP-Forward	aaagagccagacagtggtaccataacttcgtatagcatacattatacgaagttataaaaattgaaaat	1121 bp
Primer3-137-EGFP-Reverse	tacataaaaaaactatctagaataacttcgtataatgtatgctatacgaagttattaagatacattgat	1121 bp
Primer4-137-Forward	gggttagttggtagaataggaagaa	230 bp
Primer4-137-Reverse	cccagcaatcctaataacactaaag	230 bp

## Data Availability

All the relevant data are contained within this article and the Appendix A. The data that support the findings of this study are openly available in figshare at https://www.scidb.cn/s/VbAn6r (accessed on 16 September 2025).

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
