# Peer review of "Lumpy Skin Disease Virus ORF137 Protein Inhibits Type I Interferon Production by Interacting with and Decreasing the Phosphorylation of IRF3"

_cells, 2025, doi:10.3390/cells14181475_

Round 1

Reviewer 1 Report (Previous Reviewer 2)

Comments and Suggestions for Authors

The authors have addressed the comments and suggestions I made on the submitted version of their manuscript, except for those related to Figure 1.

The authors have confirmed what I suspected, that the details of the other LSDV gene products shown in Fig 1A, with the exception of ORF 137 reported in this study, will be disclosed in subsequent publications. I have no objection to this, as reporting of similar data for these other genes in the same manuscript would make it a long and cumbersome study. However, it is not appropriate for the authors to include the primary data for these genes and keep their identities confidential, as shown in the current version of the figure, as they serve no purpose in relation to the current study.

If the authors wish to retain Fig 1A as it is currently shown, the names of these genes must be added to the graph.

If the authors do not wish to disclose the identities of these genes, then they should be removed from the graph, with only the data for ORF137 shown. 

In the associated text for Fig 1A, the authors could state something along the lines of what is shown below.

Current text from Page 6 Paragraph 3:

In order to screen the viral proteins encoded by LSDV inhibiting cGAS-STING-induced IFN-β promotor activation, HEK-293T cells were transfected with 156 eukaryotic expression plasmids expressing LSDV encoding proteins and IFN-β luciferase reporter plasmids to perform dual-luciferase reporter assay. These assays identified several LSDV proteins that had significant effects on reporter gene expression. Of these, the LSDV ORF137 protein significantly inhibited the activation of IFN-βinduced by cGAS-STING to the greatest extent and was selected for further analyses in this study (Fig. 1A). The results for the remaining LSDV proteins will be reported elsewhere.

Author Response

Thank you for your comments and suggestions concerning our manuscript entitled “Lumpy skin disease virus ORF137 protein inhibits type I interferon production by interacts with and decreases the phosphorylation of IRF3”(Manuscript ID: cells-3894858).Those comments are all valuable and very helpful for revising and improving our paper, as well as the important guiding significance to our researches. We have studied comments carefully and have made correction which we hope meet with approval. The revised and annotated content has been marked with a striking green color at the corresponding positions in the manuscript.

Comments and Suggestions for Authors: The authors have addressed the comments and suggestions I made on the submitted version of their manuscript, except for those related to Figure 1.

The authors have confirmed what I suspected, that the details of the other LSDV gene products shown in Fig 1A, with the exception of ORF 137 reported in this study, will be disclosed in subsequent publications. I have no objection to this, as reporting of similar data for these other genes in the same manuscript would make it a long and cumbersome study. However, it is not appropriate for the authors to include the primary data for these genes and keep their identities confidential, as shown in the current version of the figure, as they serve no purpose in relation to the current study.

If the authors wish to retain Fig 1A as it is currently shown, the names of these genes must be added to the graph.

If the authors do not wish to disclose the identities of these genes, then they should be removed from the graph, with only the data for ORF137 shown.

In the associated text for Fig 1A, the authors could state something along the lines of what is shown below.

Current text from Page 6 Paragraph 3:

In order to screen the viral proteins encoded by LSDV inhibiting cGAS-STING-induced IFN-β promotor activation, HEK-293T cells were transfected with 156 eukaryotic expression plasmids expressing LSDV encoding proteins and IFN-β luciferase reporter plasmids to perform dual-luciferase reporter assay. These assays identified several LSDV proteins that had significant effects on reporter gene expression. Of these, the LSDV ORF137 protein significantly inhibited the activation of IFN-βinduced by cGAS-STING to the greatest extent and was selected for further analyses in this study (Fig. 1A). The results for the remaining LSDV proteins will be reported elsewhere.

Response 1: Thanks for your careful checks. Based on the reviewers' comments and suggestions, we have decided to remove the data of other LSDV genes shown in Figure 1A that are irrelevant to this study, retaining only the data of the ORF137 gene (i.e., Figure 1A has been revised). Meanwhile, we have modified the corresponding part of the text that describes this result.

Reviewer 2 Report (Previous Reviewer 1)

Comments and Suggestions for Authors

All previous comments have been addressed

Author Response

We would like to thank you for your professional review work, constructive comments, and valuable suggestions on our manuscript. 

All previous comments have been addressed.

Round 2

Reviewer 1 Report (Previous Reviewer 2)

Comments and Suggestions for Authors

The authors have addressed all of my comments and suggestions. I have no further comments to make on the current version of their manuscript.

This manuscript is a resubmission of an earlier submission. The following is a list of the peer review reports and author responses from that submission.

Round 1

Reviewer 1 Report

Comments and Suggestions for Authors

Please see attached

Reviewer 2 Report

Comments and Suggestions for Authors

The authors have investigated the capacity of lumpy skin disease virus (LSDV) to interfere with innate signalling. Using synthetic ORFs for the 156 genes of LSDV they identified ORF137 as a candidate for interfering with innate immune signalling and subsequently characterised various molecular interactions involving it.

The introduction provides the required information to understand the study and the references cited a relevant. The aims of the study are clearly articulated.

The materials and methods are described in sufficient detail to enable replication of the study.

The results are for the most part clearly illustrated, and the text describes what is depicted in the figures. However, it is not clear to me how the authors selected ORF137 as the main subject of their study. Based on the data shown in Fig 1A there appears to be several candidate genes, though they are not named. For example, genes 7, 10, 15, & 22 appear to have similar effects in comparison to ORF137. Though the impacts are not shown as being significant. Where they subject to similar statistical analyses? While other “genes” (notably 5, 9, 14, & 16) appear to have stimulatory effects. However, based on the figure, only ORF137 is shown as significant. It is not clear to me what the authors have included these additional data. That LSDV would have multiple genes have similar roles is not surprising. Why the authors elect to show these data and not report or discuss them requires some explanation.

The discussion also requires some improvement. There are new results introduced – see comments below. The purpose of the discussion is to place the study results in the context of the current literature. It is does this to some extent, but it is mixed up with new results. See comments below.

The conclusions are supported by the results of the study.

It is great the authors have provided the uncropped images of their Western blots as a supplemental file. I gather the blots were probed as cut down membranes. Is this correct? Examples of full membranes is preferable (at least an example of) otherwise it is difficult to assess specificity if only a section of the membrane is show.

Line 38 suggest replacing “homology” with “identity”

Line 44 suggest revision “in 1929 and mostly circulated in Africa, until 1986 when it emerged in Israel, gradually spreading”

Lines 114-129 – I would suggest revising this text as a table.

Line 163 suggest revision “At 24 h post-transfection, cells”

Line 194 suggest revision “At 24 h post-transfection, the cells”

Line 249 to 251 I do not understand what the message of this sentence is. I think the authors are saying that Fig 1A illustrates selected results from the double luciferase reporter studies with the LSDV ORFs.

See comments for Fig 1 below.

Line 261 suggest revision “ORF137 slightly affected”

Though I think the authors should be more quantitative in their descriptions.

Line 265 Figure 1.

The authors need to provide the names of the cryptically labelled LSDV ORFs shown in Fig 1A, the use of arbitrary names is pointless.

The figure only illustrated the detection of a significant difference for LSDV137. Can the author please confirm that none of the other illustrated results differed significantly from the comparator sample?

Similarly, the authors should state what the comparator sample is in the legend. They should also state what the columns and error bars represent and number of replicates.

The methods suggest that constructs were made for all the LSDV ORFs. If they were all used in these transfection studies, the results not illustrated here could be provided as a supplemental table.

Line 297 Figure 2

Fig 2E – suggest replacing “Green light” with “fluorescence”

Line 309-310 – Did the authors consider an earlier time point in this experiment?

Line 324 Figure 3

Fig. 3A The x-axis should have a label “Time post infection (h)”, with only the numbers shown on the graph.

Fig 3B suggest replacing “LSDV” with “Time post infection” in the top row.

Line 340 The Alphafold 3 server provides details on the specific way to acknowledge the use of their tools, see required citation on the page.

Line 370 suggest revision “To further verify the capacity of LSDV ORF137 to modulate innate immune signalling”

Line 376 suggest deletion of “obviously”

Line 370 Figure 5B – suggest replacing “red light” with “fluorescence”

Line 376 The authors have not statistically evaluated these data and as such the use of the term “significantly” should not be used.

Line 384-388 The authors should ensure that their interpretation of the data is not solely based on statistics. While the data illustrated in Fig 5F clearly illustrates significant downregulation of the genes of interest, the effects are modest, robust but modest. The authors should consider quoting the magnitude of the observed effects to provide some context for their quoting the statistical differences.

Line 434 Figure 6

Fig. 6C - suggest replacing “Green light” with “fluorescence”

Line 478 The authors should not introduce new results in the discussion. These results should be moved to the results section, and a brief description of the methods used should be added to the appropriate section.

Line 539 Figure 7 does not appear to have been included in the manuscript.

Line 546 What is the purpose of Fig. S1? It does not appear to have been cited in the main text. The other supplemental figures should also have unique identifiers.

Comments on the Quality of English Language

See comments to authors, nothing major required.